# Semi-Automating Knowledge Base Construction
# for Cancer Genetics

**Somin Wadhwa**                                    SOMINWADHWA@CS.UMASS.EDU
*College of Information & Computer Sciences*
*University of Massachusetts, Amherst*
*Amherst MA 01002, USA*

**Kanhua Yin, M.D.**                                    KYIN@MGH.HARVARD.EDU
*Division of Surgical Oncology, Massachusetts General Hospital*
*Harvard Medical School, Harvard University*
*55 Fruit St, Boston MA 02114, USA*

**Kevin S. Hughes, M.D.**                                    KSHUGHES@PARTNERS.ORG
*Division of Surgical Oncology, Massachusetts General Hospital*
*Harvard Medical School, Harvard University*
*55 Fruit St, Boston MA 02114, USA*

**Byron C. Wallace**                                    B.WALLACE@NORTHEASTERN.EDU
*Khoury College of Computer Sciences*
*Northeastern University*
*Boston MA 02115, USA*

## Abstract

The vast and rapidly expanding volume of biomedical literature makes it difficult for domain experts to keep up with the evidence. In this work, we specifically consider the exponentially growing subarea of genetics in cancer. The need to synthesize and centralize this evidence for dissemination has motivated a team of physicians to manually construct and maintain a knowledge base that distills key results reported in the literature[1]. This is a laborious process that entails reading through full-text articles to understand the study design, assess study quality, and extract the reported risk estimates associated with particular hereditary cancer genes (i.e., *penetrance*, defined as the risk of cancer with a pathogenic variant in a germline cancer susceptibility gene). In this work, we propose models to automatically surface key elements from full-text cancer genetics articles, with the ultimate aim of expediting the manual workflow currently in place.

We propose two challenging tasks that are critical for characterizing the findings reported in penetrance studies: (i) Extracting snippets of text that describe *ascertainment mechanisms*, which in turn inform whether the population studied may introduce bias owing to deviations from the target population; (ii) Extracting reported risk estimates (e.g., odds or hazard ratios) associated with specific germline mutations. The latter task may be viewed as a joint entity tagging and relation extraction problem. To train models for these tasks, we induce distant supervision over tokens and snippets in full-text articles using the manually constructed knowledge base. We propose and evaluate several model variants, including a transformer-based joint entity-relation extraction model to extract `<germline mutation, risk-estimate>` pairs for different cancer types. We observe strong empirical performance, highlighting the practical potential for such models to aid KB construction in this space. We ablate components of our model, observing, e.g., that a joint model for

---

1. https://ask2me.org/index.php

`<germline mutation, risk-estimate>` fares substantially better than a pipelined approach.

## 1. Introduction

The published evidence base concerning genetic factors in cancer is vast and growing rapidly. A simple search for "gene cancer study" on PubMed[2] yields over 238,000 research articles (including clinical trial reports, meta-analyses, and other types of research); about 16,500 of these were published in 2019 alone. This torrent of unstructured findings makes it difficult for domain experts to navigate and make sense of the evidence. Consequently, there is a critical need for tools designed to help clinicians monitor and synthesize the literature on genetics in cancer [Bao et al., 2019].

Indeed, the medical domain relies on centralization of knowledge [Collins and Varmus, 2015, Landrum et al., 2017, Forbes et al., 2017], because sorting through the vast published literature is too onerous for individual clinicians. Therefore, structured knowledge bases (KBs) derived from the literature have an important role to play in medicine generally, and in tracking progress and disseminating findings relevant to cancer genetics specifically.

Biomedical Natural Language Processing (NLP) methods provide a means of extracting key information from the scientific and biomedical literature in general [Kim, 2017, Luan et al., 2017]. And there has been some work specifically aiming to aid medical evidence synthesis via NLP techniques [Lehman et al., 2019, Marshall et al., 2018, Cohen et al., 2006, Brockmeier et al., 2019, Schmidt et al., 2020]. More specific to this work, a few efforts have focused on models for identifying literature relevant to cancer susceptibility [Wallace et al., 2012, Bao et al., 2019]. This paper extends state-of-the-art NLP technologies to aid identification and extraction of relevant information from biomedical research articles on gene-cancer associations. We envision a semi-automated process in which these models aid clinicians to facilitate efficient maintenance of a genetics in cancer knowledge base.

**The main contributions of this work are as follows**.

1. To our knowledge, this is the first effort to (semi-)automate extraction of key evidence from full-text cancer genetics papers. We introduce the tasks of extracting ascertainment mechanisms and reported risk metrics for particular mutations from the genetics in cancer literature. We propose distantly supervised strategies for learning models for one of these tasks. In contrast to most prior work, we operate over *full-text* articles rather than abstracts only; this is critical as key information will not always be found in abstracts. We envision these models *aiding* domain experts (with whom this is a collaborative effort) in maintaining a KB, rather than fully automating this process.

2. We propose and evaluate a joint model for extraction of relevant reported (numerical) measures of cancer risk and the germline mutations to which they correspond. We realize strong empirical performance using this approach, and perform ablations to investigate which components drive this.

---

2. A repository of biomedical literature: https://www.ncbi.nlm.nih.gov/pubmed/.

| PMID | Gene | Cancer | Race | OR | RR | HR | Max Age | Total Carriers |
|---|---|---|---|---|---|---|---|---|
| 29922827 | BRCA2 | Pancreatic | Multiple | 6.2 | - | - | - | 370 |
| 29922827 | TP53 | Pancreatic | Multiple | 6.7 | - | - | - | 31 |
| 27595995 | CHEK2 | Breast | White | 3.39 | - | - | 75 | 11 |
| 21145788 | MSH2 | Colorectal | Multiple | - | - | 0.49 | | - |

Table 1: Entries from the structured data currently manually curated by clinicians via manual reading of and extraction from cancer-genetics literature. This is inexhaustive, the resource also contains elements such as the carriers classified by disease-type, age-risk, paper type, and the ascertainment sentences. The idea is to design models that consume full-text articles and extract these elements.

This is joint work with specialists at the Massachusetts General Hospital (MGH) who have up to now been manually synthesizing the medical literature associated with gene-cancer associations into a KB (see Table 1 for an illustrative, condensed fragment). This is a valuable yet tedious and time-consuming endeavor. We identify targets which, if successfully extracted automatically, could greatly reduce the human labor required for upkeep of this KB. We show that by extending state-of-the-art NLP models and training these via distant supervision induced from the existing KB, we can achieve strong performance, which in turn suggests that such models may significantly reduce manual workload without compromising the ability to extract relevant information (i.e., without sacrificing the accuracy and comprehensiveness of the resultant KB).

Our effort here naturally extends prior work that used NLP to identify literature relevant to genetics in cancer [Bao et al., 2019]. However, we focus on *extraction* of key information from the full-texts of such articles, once identified. Following discussions with the specialist team and a data exploration phase, we identified two tasks corresponding to elements that were both highly relevant to the medical researchers (i.e., key elements of their KB) and potentially extractable from full-texts using natural language technologies.

1. Identify snippets that correspond to *ascertainment* of populations studied in the underlying research described in articles. Ascertainment is a complex concept; here we adopt a working definition such that snippets are considered relevant to ascertainment if they answer any of the following key questions.

   - What was the source of the study population, including geographical locations, ethnicities, and source cohorts (e.g., cancer registries, hospital-based retrospective cohorts, and community-based prospective cohorts)?

   - How many patients (cases) and/or controls were being enrolled?

   - What were the inclusion and exclusion criteria when enrolling in the study population? For example, a study may only enroll early-onset breast cancer patients, or patients with a strong family history of cancer.

   We aim to identify snippets of text that convey this information.

2. Perhaps the most important data elements in cancer in genetics studies are the risk estimates (i.e. penetrance) associated with a particular pathogenetic germline mutation variant and a specific type cancer (e.g., BRCA2 mutation) type. This information will be reported numerically using a standard metric that quantifies comparative risk, typically one of: Odds Ratio (OR), Relative Risk (RR), or Hazard Ratio (HR). These are often reported in free text and only implicitly reference the corresponding genetic variant, for example: *"odds ratio for basal cell carcinoma was higher (OR=3.8; 95% CI, P = 0.002)"*. We provide more details in Section 2.

The remainder of this paper is structured as follows: We start by providing an overview of our approach in Section 2, explaining how the data was curated (2.2) and (distantly) labeled (2.3). Section 3 and Section 4 describe our experimental setup and our results, respectively. We summarize prior related work in Section 5. Finally, we conclude in Section 6 with a discussion of the implications of this work and possible future directions in this area.

## 2. Methods

### 2.1 Overview

Given a Portable Document Format (PDF) full-text research article on gene-cancer associations, our system aims to extract sentences that describe the ascertainment mechanism for the underlying study, and spans that report the risk metrics for the specific gene-cancer association. For these tasks we define two pipelines that both operate over sentences comprising full-texts.

### 2.2 Data and Targets

Our clinical collaborators provided a risk object database (ROD) $\mathcal{D}$ comprising data that they manually extracted from 597 penetrance papers reporting gene-cancer associations, i.e., $\mathcal{D}$ contains the sought-after KB elements for these papers. The semi-structured elements in $\mathcal{D}$ were manually extracted from full-text articles, to which we also have access. The majority of these (588) are PDFs, the remaining (9) are in HTML format.

The sentences pertaining to ascertainment (as described in Section 1) within $\mathcal{D}$ were identified as key targets for extraction. We provide examples of these below. In addition to sentences describing what demographic population was studied, ascertainment sentences included those indicating "adjustments" for ascertainment. For example, the following sentences were distantly labeled as positive for ascertainment.

- A hospital-based study or a panel testing analysis with well-matched cases and controls, such as:

  1. *A control population was defined from the National Danish Civil Registration System, with five population controls, matched on sex and year of birth, for mutation carriers as well as first-degree relatives.*

  2. *For age-adjusted analysis, the projected U.S. population (year 2000) was used; 84% of the 3,399 individuals were white.*

| Text | Targets |
|---|---|
| These included **CDKN2A**, with mutations in 0.30% of cases and 0.02% of controls (OR, **12.33**; 95% CI, 5.43-25.61); | |
| **TP53**, with mutations in 0.20% of cases and 0.02% of controls (OR, **6.70**; 95% CI, 2.52-14.95); | <CDKN2A, 12.33, positive> <TP53, 6.70, positive> |
| **MLH1**, with mutations in 0.13% of cases and 0.02% of controls (OR, **6.66**; 95% CI, 1.94-17.53); | <MLH1, 6.66, positive> <BRCA2, 6.20, positive> |
| **BRCA2**, with mutations in 1.90% of cases and 0.30% of controls (OR, **6.20**; 95% CI, 4.62- 8.17); | <BRCA2, 4.62, negative> <CDKN2A, 6.70, negative> |
| **ATM**, with mutations in 2.30% of cases and 0.37% of controls (OR, **5.71**; 95% CI, 4.38-7.33); | |

Table 2: Example of a snippet reporting risks associated with germline variants (left) and corresponding extraction or relation targets (right).

- A proband-based/family-based study with appropriate ascertainment adjustment: such as GRL, modified segregation analysis, for example:

  1. *To adjust for ascertainment, we used an ascertainment assumption–free approach in which we evaluated each family separately*

  2. *Once we verified that the SIR estimates were not influenced by such cohort-effects, our final analyses were based on population rates specific for each country, sex, and 5-year age group averaged from 1950 to 2009 which were applied to all follow-up, regardless of calendar year.*

The second key elements that we aim to extract are the reported risks associated with particular germline mutations. These are typically provided as Odds, Risk, or Hazard Ratios (ORs, RRs, HRs), i.e., floating point numeric values. These values are arguably the main results being conveyed by the article. There can be multiple metrics reported throughout the entire article, corresponding to different gene-cancer associations (see Table 2). Models must therefore exploit context to disambiguate to which gene a given metric corresponds. We derive data about these risk estimates through direct supervision as they appear in the risk object database provided to us by the medical experts.

## 2.3 Deriving Distant Supervision for Ascertainment Classification

Rather than direct supervision (explicit labels on snippets of text extracted from PDFs), we have access only to the database manually compiled by physicians thus far ($\mathcal{D}$), which comprises semi-structured elements, pertaining to ascertainment, extracted from a set of articles. We aim to induce distant supervision over article snippets by heuristically matching these to entries in $\mathcal{D}$.

To this end, we retrieved the 597 full-texts (mostly PDFs) associated with all entries in $\mathcal{D}$. We processed PDFs using Grobid[3] to generate structured XML documents. These XML encodings are reasonably structured, and we were able to compose simple rules to extract all article text (but not tables and figures).

Next, we extracted sentences from all article sections (save for the abstract) using `sciSpacy`.[4] We then constructed three different representations of these sentences:

1. **Simple Bag-of-Words**. Average of vector representations obtained via `sciSpacy`, for every token in a given sentence. We fix the embedding size to 300.

2. **Weighted TF-IDF**. Weighted average of token vectors with their respective TF-IDF scores [Schmidt, 2019].

3. **Contextual Representation**. We pass sentence tokens through SciBERT [Beltagy et al., 2019] (this is a variant of BERT [Devlin et al., 2018] pre-trained on scientific corpora) and take as a sentence representation the 768-dimensional `[CLS]` token embedding.

Once these sentence representations are obtained we generate 'positive' labels for sentences corresponding to descriptions of ascertainment mechanisms. To derive these pseudo-labels we compare all sentence representations from a document $D_i$, to its true (manually extracted) ascertainment mechanism snippet, $A_i$, using cosine distance:

$$\cos(\mathbf{s}, \mathbf{A_i}) = \frac{\mathbf{s}\mathbf{A_i}}{\|\mathbf{t}\|\|\mathbf{A_i}\|} = \frac{\sum_{j=1}^{n} \mathbf{s}_j \mathbf{A_{i}}_j}{\sqrt{\sum_{j=1}^{n} (\mathbf{s}_j)^2}\sqrt{\sum_{j=1}^{n} (\mathbf{A_i}_j)^2}} \tag{1}$$

Extracted ascertainment mechanisms often correspond to more than one sentence in a document (on average, there are 2.6 of these per document). Therefore, we retain the top three sentences with the highest similarity scores, and we mark these as 'positives'. One issue with this approach is that using a similarity measure often yields 'false positives' that are more general than what we are after. We discus this further in the Appendix.

## 2.4 Joint Entity-Relation Extraction to Determine Risk Estimates

A key piece of information reported in genetics in medicine studies is the reported risk ratio for particular germline mutations. In the risk object database ($\mathcal{D}$), a given row corresponds to one cancer risk estimate for a specific `<germline mutation, risk-estimate>` pair and one penetrance paper may correspond to multiple such rows in $\mathcal{D}$. Therefore, our second task is to extract these metrics (Odds/Risk/Hazard Ratio) and identify the germline mutation to which they correspond. We perform this extraction over individual sentences independently.

Figure 1 provides a schematic of our approach. For this task we propose a transformer-based model that jointly extracts the relevant entities (risk-estimates, genes) and predicts their relations (positive, negative) directly from spans within individual sentences.

---

3. Grobid is a tool for extracting and parsing PDFs into structured XML/TEI encoded documents with a particular focus on technical and scientific publications: https://github.com/kermitt2/grobid-client-python.

4. A sister NLP library to `SpaCy` (https://spacy.io/) built for biomedical text: https://allenai.github.io/scispacy/

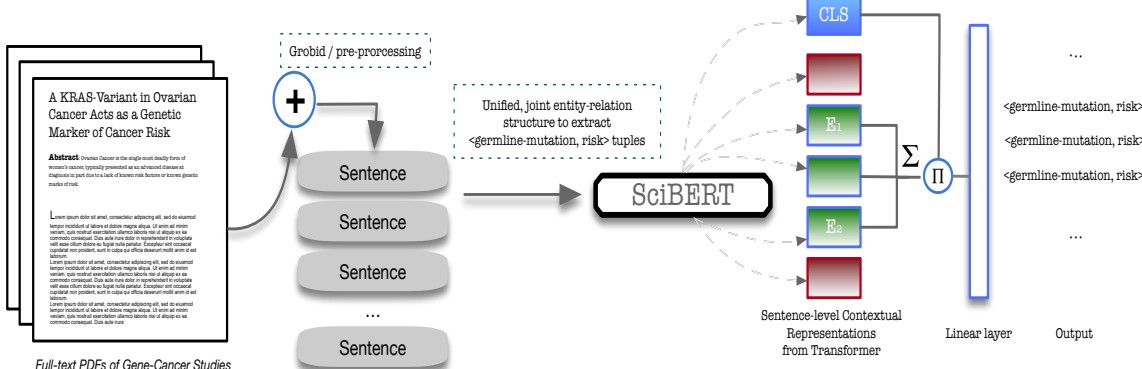

Figure 1: A schematic of our joint model for extracting germline-mutations and corresponding risk metrics. See text for additional description.

Concretely, an input document $X$ is treated as a set of sentences $\mathcal{X} = s_1, s_2, s_3, \ldots, s_{|X|}$. The task of entity recognition entails predicting a label $e_i$ (here: `germline mutation`, `risk-estimate`, or `none`) for each token within a sentence. Relation extraction involves predicting whether a `<germline mutation, risk-estimate>` pair is a positive (true) relation or not.

Our model considers fixed length (10-15) spans within sentences and classifies each token into possible entity types. The model then classifies these entity-types into relations (*positive* or *negative*).

We use SciBERT [Beltagy et al., 2019] to generate token level representations. We sum these and the `[CLS]` token embedding to incorporate overall context, yielding the final input for entity classification:

$$e_i = \text{softmax}(W_e \cdot ([\texttt{CLS}] + t_i)) + b_e) \tag{2}$$

For every pair of entity candidates $(e_x, e_y)$, where neither is of type `none`, our next task involves predicting the best relation type $r_{x,y}$. To form a combined localized representation of $(e_x, e_y)$, we take the sum of the embeddings from $e_x$ through $e_y$ and then take the element-wise product of this representation with the `[CLS]` token representing the span. We pass this output through a fully connected layer followed by a sigmoid, yielding a binary probability prediction. Therefore, if $i$ and $j$ correspond to the token indices of entities $e_x$ and $e_y$ respectively, then the predicted probability corresponding to $r_{x,y}$ is:

$$r_{x,y} = \sigma(W_r \cdot ([\texttt{CLS}] \cdot \sum_{k=i}^{j} (e_k) + b_r) \tag{3}$$

Much like [Luan et al., 2019], our loss function is the (unweighted) sum of the log-likelihood of the two tasks:

$$\sum_{(D,R*,E*)\epsilon\mathcal{X}} \{\log \mathcal{P}(R^*|E, D) + \log \mathcal{P}(E^*|D)\} \tag{4}$$

Where $E^*$ and $R^*$ are the true entities and relations, respectively. $\mathcal{X}$ is the collection of all training documents.

**Simple, disjoint model for entity-relation extraction:** Does joint training yield improved performance? To investigate this we also train two simple disjoint models, based on [Shi and Lin, 2019], for the same task. In this approach, entity tokens are classified using a dense layer on top of representations induced by SciBERT [Beltagy et al., 2019]. Then, for the relation extraction task, the model takes input as [[CLS] `sentence-span` [SEP] `entity` [SEP] `entity` [SEP]]. We then discard the sentence after the first [SEP] token, preserving only the contextualized representation which is then concatenated and passed into a fully connected network to obtain the output.

## 3. Experimental Setup

We use `sciSpacy` [Neumann et al., 2019] to carry out all text preprocessing, including tokenization and sentence splitting. To train our transformer based models, we use the HuggingFace [Wolf et al., 2019] transformer library in PyTorch [Paszke et al., 2019]. We're providing the model code along with the raw annotations of penetrance papers, indexed by their PubMed IDs[5].

| Basic Data Statistics | | | | | | | | |
|---|---|---|---|---|---|---|---|---|
| | **Train** | **Val** | **Test\*** | | | **Train** | **Val** | **Test** |
| Positive | 10414 | 1838 | 612 | Entities (g/r.e.) | | 2134/1549 | 267/194 | 267/193 |
| Negative | 55920 | 9866 | 6589 | Entity-Relation Pairs | | 1549 | 194 | 193 |

Table 3: g/r.e corresponds to genes/risk-estimates.

**Data**: We obtained annotated data (Table 3) from clinicians as described in Section 2.2. Table 3 reports basic data statistics and information about train, development, and test set splits that we have created. Importantly, for both of our tasks, these are disjoint at the *document level*, meaning that sentences in the respective splits are drawn from corresponding unique sets of documents.

**Training**: For the ascertainment classification task, our best performing model on dev data (fine-tuned, SciBERT) uses the Adam [Kingma and Ba, 2014] optimization scheme, a batch size of 32 and a learning rate of 2e-5 over 4 epochs.

For the joint entity tagging and relation extraction task, we again use the Adam optimizer, with a batch size of 16, and a linear decay learning rate schedule. The learning rate here is 5e-5. We observe that the model peaks after 8-10 epochs. We also observe that the same hyperparameters suffice with our disjoint entity-relation classification models.

All of our hyperparameter tuning was carried out on development sets; the results we report are from the held out test set, unseen during model development.

| Model | F1 | P | R | MCC | Acc |
|---|---|---|---|---|---|
| SVM (word2vec, BOW) | 0.71 | 0.81 | 0.68 | 0.46 | 0.83 |
| SVM (word2vec, weighted tf-idf) | 0.75 | 0.78 | 0.73 | 0.51 | 0.84 |
| Logistic Regression (word2vec, BOW) | 0.68 | 0.72 | 0.65 | 0.39 | 0.78 |
| Logistic Regression (word2vec, weighted tf-idf) | 0.72 | 0.74 | 0.71 | 0.41 | 0.78 |
| BERT (fine-tuned, uncased) | 0.80 | 0.79 | 0.82 | 0.70 | 0.84 |
| SciBERT (fine-tuned, scivocab-uncased) | **0.89** | **0.92** | **0.86** | **0.86** | **0.97** |

Table 4: Results for the weakly supervised models on the ascertainment classification task. Note that the *test* data on which we evaluate is directly labeled.

| Model | Entity | | | | Relation | | | |
|---|---|---|---|---|---|---|---|---|
| | F1 | P | R | MCC | F1 | P | R | MCC |
| SVM (word2vec, d=300) | 0.67 | 0.71 | 0.60 | - | - | - | - | - |
| disjoint entity-relation extraction | | | | | | | | |
| BERT (uncased) | 0.78 | 0.85 | 0.72 | **0.61** | 0.44 | 0.47 | 0.41 | 0.28 |
| SciBERT (scivocab, uncased) | **0.79** | **0.88** | 0.71 | 0.59 | 0.49 | 0.56 | 0.43 | 0.29 |
| joint entity-relation extraction | | | | | | | | |
| BERT (uncased) | 0.77 | 0.85 | 0.69 | 0.59 | 0.61 | 0.68 | 0.54 | 0.33 |
| SciBERT (scivocab, uncased) | 0.78 | 0.83 | **0.74** | 0.59 | **0.62** | **0.70** | **0.58** | **0.35** |

Table 5: Test set results on entity-relation extraction task

## 4. Results

We report results on test sets in Tables 4 and 5. For all models, we report F1 score, precision, recall, accuracy, and matthews correlation coefficient (MCC).

For the first task of ascertainment classification, we observe that a transformer based approach (i.e., a simple classification layer on top of SciBERT) outperforms baselines, as one would expect. However, we were somewhat surprised by the performance difference observed between variants initialized with BERT and SciBERT weights; the latter achieves a 9 point absolute gain in F1, apparently demonstrating the considerable advantage afforded by domain-specific pre-training.

For the second task of extracting `<germline mutation, risk-estimate>` pairs, we observe similar performance (using SciBERT) on the entity classification task with both joint and the pipelined approach. However, the joint model outperforms the pipelined approach in the subsequent relation inference task, yielding a substantial difference of 13 points in F1 score. This demonstrates the promise of joint — as opposed to pipelined — approaches for this task.

Qualitatively, we observe that our models produce high quality results, in the sense that the ostensibly misclassified examples are ambiguous to a clinical evaluator. We provide

---

5. https://github.com/sominwadhwa/kbcCanGen.

evidence for this along with additional ablation experiments (e.g., model sensitivity to specific numerical tokens for risk ratio extraction) in the Appendix.

## 5. Related Work

Work related to this effort includes general methods for scientific information extraction [Liu et al., 2016, Luan et al., 2019], evidence mining [Rinott et al., 2015], and the use of distant supervision to create weakly (and noisily) labelled data at scale [Mintz et al., 2009, Augenstein et al., 2015].

Recent work on exploiting distant supervision has focused largely on the analysis of free text on social media [Purver and Battersby, 2012, Marchetti-Bowick and Chambers, 2012] and on relation extraction [Riedel et al., 2010, Mintz et al., 2009, Nguyen and Moschitti, 2011]. Our work relates more to the former approaches of using heuristic and distant supervision to induce weak, sometimes noisy, labels for a downstream task. Despite this inherent noise, we find that we are able to exploit such weak labels to train layers on top of modern contextualized encoder models [Devlin et al., 2018, Peters et al., 2018] pretrained on scientific text [Beltagy et al., 2019], yielding extractors that can capture the desired information with high fidelity. This extends past work in which weak or distant forms of supervision have specifically been used to train models for information extraction from scientific and biomedical texts [Quirk and Poon, 2016, Jain et al., 2016, Wallace et al., 2016, Norman et al., 2019].

Designing language technologies for (semi-)automated information extraction in the biomedical domain remains an active area of research [Liu et al., 2016, Patel et al., 2018]. For example, there is a line of related work designing and evaluating extraction methods to aid biomedical literature synthesis [Jonnalagadda and Petitti, 2013, Kiritchenko et al., 2010, Nye et al., 2018]. Much of the recent work in this space has involved using neural models to extract entities [Greenberg et al., 2018], relations between these entities [Song et al., 2018, Krasakis et al., Verga et al., 2018, Wadden et al., 2019] and reported findings [Lehman et al., 2019]. We build on these models in the present work.

In oncology research specifically, classification models have been applied to designate staging and assess cancer recurrence [Deng et al., 2019, Hughes et al., 2020, Friedlin et al., 2010]. Fiszman et al. [2010] used semantic analysis of text to identify cardiovascular risk factors in medical literature. Elsewhere, Lossio-Ventura et al. [2016] explicitly considered the long-term aim of constructing an obesity-cancer KB, though their work was limited to studies in obesity and titles/abstracts, rather than full-texts.

None of the aforementioned works have attempted to infer population characteristics or risk estimates from the cancer in genetics literature, despite the importance of establishing a centralized KB for the cancer in genetics literature, and the burden that manually curating and maintaining this resource currently imposes. This effort is a first step toward building models to mitigate manual workload, thereby allowing KB construction to scale as the literature continues to rapidly grow.

## 6. Conclusions

We have proposed the practically important task of extracting relevant information from cancer genetics literature, with the aim of helping domain experts (clinicians) maintain an up-to-date knowledge base of genetics in cancer results.

We acquired full-text PDFs of gene-cancer studies from which clinicians previously extracted key structured information from full-texts into a database, and we used these entries to induce distant supervision over spans within the corresponding articles. Using these derived (weak) labels, we trained a classification model to identify key snippets that relate to ascertainment bias, a key consideration in such evidence.

We then considered the more challenging task of extracting `<germline-mutation, risk>` pairs as an entity-relation problem. For this we proposed a BERT-based joint entity tagging and relation extraction model. Through ablation we observe: (1) This joint approach fares substantially better than a two-step pipelined approach, and, (ii) Initialization to BERT parameters learned "in-domain" also provides a considerable performance increase [Beltagy et al., 2019].

Going forward, we hope to evaluate the utility of these models *in practice* to assess the degree to which they actually help expedite knowledge-base construction. This work will assist the domain experts in: assessing study quality quickly and accurately, identify the most representative population-level risk for a specific gene-cancer association. Ultimately, risks identified in the primary literature may serve as a foundation for providing individualized cancer prevention practice and clinical care. More generally, we hope this first effort spurs additional work on models for automatically making sense of the genetics in cancer literature.

## 7. Acknowledgements

This work was funded in part by the National Institutes of Health (NIH) under the National Library of Medicine (NLM) grant 2R01LM012086, and by the National Science Foundation (NSF) CAREER award 1750978.

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

## Appendix

To evaluate the effectiveness of our models, we analyse a range of outputs at the inference level to gauge what our model learns, and to assess its usability with respect to variations in input data.

For ascertainment classification, we inspect some of the misclassified sentences in the test set to highlight their commonalities, and to better understand what our models learn.

- Example false positives:

    - Only 129 unique eligible studies met our **criteria** for **inclusion** and had sufficient data available for extraction.

    - However, after sequencing a **larger cohort of patients** in those populations (figure 2), a significant enrichment in these populations was observed among cases.

    - **Ethnicity** is usually based on 80% of the **study population**, and if not reported, we considered **ethnicity** as the country of publication.

    - Male patients with breast or renal cancer had an increased prevalence of thyroid caner of 19-and three-fold, respectively.

    - This step yielded 560 studies containing 66 snvs in 51 different genes that were eligible for the **inclusion criteria** in this study.

- Example false negatives:

    - Families fulfilling amsterdam ii criteria with normal expression of mmr proteins or microsatellite stable tumors were considered as familial crc type x, and genetic analysis of pole and pold1 was performed.

    - Finally, the references of all studies included were scanned, as were reference lists from relevant reviews and meta-analyses.

    - All proven or obligate mmr gene mutation carriers from the hnpcc register were eligible for the study and are referred to as the lynch syndrome cohort.

    - We chose a threshold of at least 4 studies before performing a meta-analysis for subgroups.

    - Forty-five of the patients diagnosed with malignancy possessed mutations that result in truncation of the expressed protein.

We often observe that false positives include certain aforementioned keywords that frequently appear in true ascertainment sentences. For example, in the ascertainment classification task, false positive examples often include the words *population, registry, demographic, inclusion criteria*, and so on. This indicates that even when the sentence is mislabelled, the prediction still relates to the original concepts.

On our second task of extracting entity-relation pairs, we perform an ablation experiment to test the effect of contextualized representations of spans while jointly classifying entities and their relations. We primarily define three tasks according the way we alter our test data.

| Task | Entity | | | | Relation | | | |
|------|------|------|------|------|------|------|------|------|
| | F1 | P | R | MCC | F1 | P | R | MCC |
| A | 0.74 | 0.82 | 0.69 | 0.55 | 0.60 | 0.67 | 0.54 | 0.30 |
| B | 0.77 | 0.83 | 0.72 | 0.57 | 0.62 | 0.69 | 0.57 | 0.33 |
| C | 0.68 | 0.76 | 0.64 | 0.39 | 0.51 | 0.55 | 0.48 | 0.28 |

Table 6: Test set results for fuzzy inputs at inference time on the ER extraction task with the joint transformer based model (SciBERT).

- **Task A:** In the spans containing risk-estimate values (floating numerics), we increase the value of all numeric tokens by an order of $10^3$.

  - Example: These included **CDKN2A**, with mutations in 0.30% of cases and 0.02% of controls (OR, **12330**; 95% CI, 5430-25610); True `<germline-mutation, cancer>`, in this case corresponds to, `<CDKN2A, 12330>`.

- **Task B:** In the spans containing risk-estimate values (floating numerics), we decrease the value of all numeric tokens by an order of $10^3$.

  - Example: These included **CDKN2A**, with mutations in 0.30% of cases and 0.02% of controls (OR, **0.01233**; 95% CI, 0.00543-0.02561); True `<germline-mutation, cancer>`, in this case corresponds to, `<CDKN2A, 0.01233>`.

- **Task C:** In the spans containing risk-estimate values (floating numerics), replace some true risk estimates with random non-numeric tokens.

  - Example: These included **CDKN2A**, with mutations in 0.30% of cases and 0.02% of controls (OR, **XYZ**; 95% CI, 5430-25610); True `<germline-mutation, cancer>`, in this case corresponds to, `<CDKN2A, XYZ>`.

Table 6 summarizes our results on these three tasks. While for Task B, results remains relatively stable, we observe a drop in performance for Task C.