# OpenReview forum: "Semi-Automating Knowledge Base Construction for Cancer Genetics"
_AKBC.ws/2020/Conference — AKBC 2020_

### Official Review · AnonReviewer1 · 2020-03-28
**Meaningful application, good method design, and comprehensive experiments, but would like to see a test on unlabeled data.**

**Rating:** 7
**Confidence:** 4

**Review:**

This paper applies the state-of-art NLP techniques to (semi-) automate information extraction for cancer genetics.

Quality: Good

Clarity: Pretty clear and easy to read

Originality: The authors claim that this the first effort to (semi-) automate the extraction of key evidence from full-text can generics papers.

Significance of this work: Its significance lies in the application of NLP techniques for cancer genetics KBC.

Pros:
1.	The potential application is meaningful, which will help physicians to construct and maintain a cancer genetics KB.
2.	The designed methods and evaluations are reasonable. For extracting snippets of ascertainment text, they generate noisy labels for each sentence from human extracted snippets, and thus, convert this task to a classification task; for extracting risk estimation of germline mutation, they propose a joint model to extract gene and OR entities from each sentence and connect them by predicting each two of them has positive or negative relation.
3.	The experiments are comprehensive and the results are good. They show that their methods surpass some baselines and achieve the best performance. Some examples and an ablation study are included in the Appendix.
4.	Comprehensive literature review
5.	Good writing and clear delivery

Cons:
1.	The potential application has not been really tested yet. Even though they show that their methods perform very well on their human-labeled dataset, I would like to see the real usage of these methods on the unlabeled data. I would suggest authors to apply their methods to construct a KB from other cancer genetics papers and do a human evaluation to see the real-world usage of these methods. Or, at least include some extracted information from unlabeled data in the appendix.
2.	Some details are missing:
a.	 to derive the labels for ascertainment classification, the author defines three types of sentence representations, are they combined to compute the cos similarity?
b.	 The author mentioned the “false positives” for this labeling method, I would like to see some solutions to this problem.
c.	What is the matthews correlation coefficient (MCC)? Why do you want to use it besides F1, P, R?

---

> ### Author Response · Authors · 2020-04-08
> **Response to reviewer**
>
> Thank you for your encouraging remarks and detailed comments. A few clarifications from our end:
>
> 1. Regarding sentence representations: The three types of sentence representations reported in section 3.3 are evaluated independently of each other. We compute cosine similarity using all three of them and report results in table 4. We’ll clarify this further in the revised manuscript.
> 2. MCC is generally regarded as a balanced measure which can be used even if the classes are of very different sizes (like in case of the ascertainment classification task). It is simply another metric we report in addition to F, P, R.
>
> 3. Real world evaluation: We agree that this would be ideal, but human evaluation in practice will entail a long-term effort that integrates the models into practice over a relatively long period; we hope to perform such an evaluation eventually, but believe the current evaluations suffice to demonstrate the promise of the approach. We would like to point out that all of the examples in Appendix are in fact from an unseen test set, derived from a set of unused (during training/val time) documents.

---

### Official Review · AnonReviewer2 · 2020-03-30
**Successful application of BERT-based models to new tasks, but with limited novelty in the method**

**Rating:** 6
**Confidence:** 4

**Review:**

This work address KB construction in biomedical domain. Specifically, it proposes two tasks in cancer genetics domain: (1) extracting text snippets about ascertainment; (2) extract reported risk estimates for different germline mutations. They first created distant supervision based on existing manually extracted KB. For (1), they showed that classifier using BERT (especially SciBERT) based sentence representation significantly outperforms baseline models; for (2), they used a simple combination of BERT token emebeddings and a dense layer to jointly learn to classify spans into entities and their relation type, which performs better than SVM baselines, and disjoint learning baselines.

Strength:

1. This paper proposes two tasks with real world applications, and prepared reasonable size datasets.

2. The paper proposed models based on BERT variants and significantly outperforms simple baselines.

Weakness:

1. The novelty in the method is limited because the techniques used is straightforward combination of existing approaches, for example, sciBERT, using sum of token representation as candidate entity representations, etc.

2. There isn't many new insights about the methods. The advantage of joint learning and advantage of sciBERT vs BERT seem not surprising. The paper could benefit from more error analysis of the BERT-based models, or comparing more variants of how to use the BERT token representation (for example, how to combine them into entity representations), which can help the readers understand better the weakness of the current methods and potential directions for improvement.

Questions:

Since the ground truth is created using the distant supervision, which is imperfect, for example, the paper pointed out that there's many false positives. How do you ensure the evaluation is not influenced by the errors in the distant supervision?

typos:

abstract:
"We propose two challenging tasks that are critical for characterizing the findings reported cancer genetics studies" ==> "...reported in cancer genetics..."

page 5 bottom:
"There can multiple metrics reported throughout the ..." ==> "There can be multiple..."

---

> ### Author Response · Authors · 2020-04-08
> **Response to reviewer**
>
> Thank you for your comments on our work, we appreciate the considered feedback. We agree that the primary contribution here is not a novel method --- although we do adopt and extend state-of-the-art methods --- however, we feel that the contribution of bringing these components together and evaluating joint, BERT-based extraction models on a meaningful, novel KB task, and showing that this can be trained via weak/distant supervision, is an in-scope and useful contribution for AKBC.
>
> To your question: It is reasonable to assume that distant supervision might induce some errors, and in general is imperfect compared to direct supervision. The performance of our models on the test set  (derived from a set of unused/new documents) however seems to indicate that DS does not significantly affect the overall quality of the results. Furthermore, we assess a set of misclassified examples (false positives/negatives), specified in the Appendix section. Those examples do indeed indicate that while some sentences might be misclassified, the prediction still relates to the original concepts.
>
> Typos: Thanks for pointing those out. We’ll fix them in the revised manuscript.

---

### Official Review · AnonReviewer4 · 2020-03-30
**An application of existing techniques to biomedical IE task**

**Rating:** 6
**Confidence:** 4

**Review:**

This paper applies modern deep NLP methods (especially transformers) to information extraction from biomedical texts - specifically those dealing with cancer genomics. The task is to extract two pieces of information - sentences that describe ascertainment and entity/relation extraction for risk ratios. To perform this, the authors use distant supervision from manually curated KB. For ascertainment supervision, authors go into some detail, although I have no idea how supervision for relation extraction is derived. The manually curated KB seem to have standardised names for gene mutations which may not occur exactly in the document. How are they matched to entity mentions in the document itself ?

Data release : Do the authors intend to release the data which may arguably the most important part of this paper ?

Clarity : The section describing joint ER model needs to be re-written. The authors make token level decision for entity classification. How is this used to extract actual entities which may be multi-token ? Is it possible that a sentence has more then 2 entities (biomedical text are infamous for long sentences). If each token is classified, what is role of enumerating spans ? Why not use CRF for example ?

For the relation extraction part, why is only context between two entities considered (and not the words on either side of them) in equation 3 ?

In the disjoint model, what do we mean by discarding the sentence ? And what exactly are we concatenating ? Why not just use [CLS] token embedding for classification ?

Evaluation of DS : Can you provide any evaluation of the efficacy of the distant supervision ? In general, how many false positives occur during matching ? Also how was distant supervision generated for Entity/relation extraction part.

Cross sentence : Can you comment on how much information might be missed if we only do entity/relation extraction within a sentence ? Are there relations that may be extracted by only considering information across sentences ?

Loss of Info : How much information is lost by not considering tables ? Are there risk ratios never reported in text ? How prevalent are they ?

In general, I believe it is an interesting application paper that show distant supervision can be employed reasonably well in biomedical domain. But the writing leaves one a bit confused about the exact methodology.

---

> ### Author Response · Authors · 2020-04-08
> **Response to reviewer**
>
> Thank you for your careful comments on our work, we appreciate your time and effort. Many of the issues you raise concern over the clarity of presentation: We agree that presentation of the work can be improved, and we believe that we can adequately do so for a camera-ready version of the paper, should it be accepted. We address your points individually below.
>
> 1. Data Release: Yes! We will release all manual annotations provided to us by physicians (as a CSV). Full text articles can be accessed via their pubmed ids, which we will share. However one would (unfortunately) require some form of institutional access to download the subset of these that are behind a paywall (this of course is beyond our control). We will point to a repository comprising the annotations, model code to work with this, and documentation in the camera-ready version of this paper, should it be accepted. We agree that this is an important contribution of the work.
>
> 2. Clarity: We will rewrite this section to clarify these points. Briefly:
> The entities we’re specifically looking at here are the names of the genes (single token, e.g. brca1, chek2, brca2 etc) and numeric risk estimates (OR=6.5, RR=6.6 etc) which are identified by matching them directly to their corresponding annotation.
> After identifying entities within each span (and yes there can be more than two), each pair (name, numeric qty) is checked for a potential relationship. Enumerating this way allows us to go over the entire document efficiently, avoiding unusually long sentences (like you mentioned).
>
> 3. Relation extraction part: In Equation 3, we do in fact model context level information by incorporating the [CLS] vector. There is, however, an emphasis on context between the two entities (localized context) among which we are trying to determine a relationship.
>
> 4. Disjoint model: Section 3.4 (last para): we do not discard the sentence itself, contextual representation is indeed preserved. We simply discard the additional entity representations concatenated in the previous step. This model was trained to benchmark the performance for our joint model. We will try and convey this more effectively in the revised manuscript.
>
> 5. Regarding DS: We apologise for the confusion, but we do not really use distant supervision for the entity-relation extraction task. These quantities are derived from the manual annotations provided to us by physicians. DS is only utilized for the first task of ascertainment classification (specified in table 1, section 3.2 para 2, section 3.3 para 1). We will be more explicit about this in the writing.
>
> 6. Loss of info: This is a valid and interesting observation, and could be a potential extension of our work. However, the scope of our work was to extract information from plain text over the full length of these scientific articles. Cross sentence referencing (over very large documents) and extraction from tables would probably require fundamentally different methods, which was not our focus here. We do however hope that our first effort in this domain spurs up additional work in the area. Furthermore, we strictly consider penetrance papers only (meaning all of them should contain the risk estimates information). We will integrate a discussion of this into the revised manuscript.
>
> 7. Finally, we would like to clarify that standardized (formal) names for genes do indeed appear exactly in the document, especially in context of reporting risk estimates. As an example, we could consider BRCA (breast cancer type susceptibility protein, tumor suppressor gene family), commonly called the “caretaker gene”. However, BRCA1 and BRCA2 are unrelated proteins, and have different risk estimates reported for different populations, therefore the use of colloquial names in formal scientific text is practically non-existent. Additionally, we are also provided with a look-up table that matches all the common names to their respective genes. And these entity mentions (estimates) are matched directly via the annotations that are provided to us.

---

> > ### Comment · AnonReviewer4 · 2020-04-16
> > **Scores updated.**
> >
> > I have updated my score to 6 conditioned on authors making the necessary changes to make the paper more clear.

---

### Decision · Program_Chairs · 2020-04-30

**Decision:**

Accept

**Comment:**

The paper addresses the novel task of information extraction from cancer genomics. The reviewers have applauded the important and meaningful application area, and the comprehensive experimental design that beats state of the art. The approaches are straightforward combination of existing methods. There are also some clarity issues, which we expect authors to fix in the final version.